# Maternal obesity and metabolic disorders associate with congenital heart defects in the offspring: A systematic review

Gitte Hedermann[1]*, Paula L. Hedley[1], Ida N. Thagaard[1,2], Lone Krebs[3,4], Charlotte Kvist Ekelund[4,5], Thorkild I. A. Sørensen[6,7], Michael Christiansen[1,8]

1 Department for Congenital Disorders, Danish National Biobank and Biomarkers, Statens Serum Institut, Copenhagen, Denmark, 2 Department of Obstetrics and Gynaecology, Copenhagen University Hospital, Slagelse Hospital, Slagelse, Denmark, 3 Department of Obstetrics and Gynaecology, Copenhagen University Hospital, Amager and Hvidovre Hospital, Copenhagen, Denmark, 4 Department of Clinical Medicine, University of Copenhagen, Copenhagen, Denmark, 5 Center of Fetal Medicine, Department of Obstetrics, Copenhagen University Hospital, Rigshospitalet, Copenhagen, Denmark, 6 Department of Public Health, Section of Epidemiology, University of Copenhagen, Copenhagen, Denmark, 7 The Novo Nordisk Foundation Center for Basic Metabolic Research, Faculty of Health and Medical Sciences, University of Copenhagen, Copenhagen, Denmark, 8 Department of Biomedical Sciences, University of Copenhagen, Copenhagen, Denmark

* gihc@ssi.dk

**Data Availability Statement:** All relevant data are within the manuscript and its Supporting Information files.

## Abstract

### Background

Congenital heart defects (CHDs) are the most common congenital malformations. The aetiology of CHDs is complex. Large cohort studies and systematic reviews and meta-analyses based on these have reported an association between higher risk of CHDs in the offspring and individual maternal metabolic disorders such as obesity, diabetes, hypertension, and preeclampsia, all conditions that can be related to insulin resistance or hyperglycaemia. However, the clinical reality is that these conditions often occur simultaneously. The aim of this review is, in consequence, both to evaluate the existing evidence on the association between maternal metabolic disorders, defined as obesity, diabetes, hypertension, pre-eclampsia, dyslipidaemia and CHDs in the offspring, as well as the significance of combinations, such as metabolic syndrome, as risk factors.

### Methods

A systematic literature search of papers published between January 1, 1990 and January 14, 2021 was conducted using PubMed and Embase. Studies were eligible if they were published in English and were case-control or cohort studies. The exposures of interest were maternal overweight or obesity, hypertension, preeclampsia, diabetes, dyslipidaemia, and/or metabolic syndrome, and the outcome of interest was CHDs in the offspring. Furthermore, the studies were included according to a quality assessment score.

### Results

Of the 2,250 identified studies, 32 qualified for inclusion. All but one study investigated only the individual metabolic disorders. Some disorders (obesity, gestational diabetes, and

**Funding:** GH: 18-R109-A5193-26043, The Danish Children Heart Foundation, https://boernehjertefonden.dk, The funders had no role in study design, data collection and analysis, decision to publish, or preparation of the manuscript. 19-L-0096, The A.P. Moller Foundation, https://www.apmollerfonde.dk, The funders had no role in study design, data collection and analysis, decision to publish, or preparation of the manuscript. 19-10-0493, Aase and Ejnar Danielsen's Foundation, https://danielsensfond.dk, The funders had no role in study design, data collection and analysis, decision to publish, or preparation of the manuscript.

**Competing interests:** The authors have declared that no competing interests exist.

**Abbreviations:** CHDs, congenital heart defects; MetS, metabolic syndrome; PE, preeclampsia; BMI, body mass index; GDM, gestational diabetes; DM2, diabetes type 2; DM1, diabetes type 1; ICD-10, International Classification of Diseases, 10th revision; OR, odds ratio; RR, risk ratio; PR, prevalence ratio; CI, 95% confidence interval; PGDM, pre-gestational diabetes.

hypertension) increased risk of CHDs marginally whereas pre-gestational diabetes and early-onset preeclampsia were strongly associated with CHDs, without consistent differences between CHD subtypes. A single study suggested a possible additive effect of maternal obesity and gestational diabetes.

## Conclusions

Future studies of the role of aberrations of the glucose-insulin homeostasis in the common aetiology and mechanisms of metabolic disorders, present during pregnancy, and their association, both as single conditions and–particularly–in combination, with CHDs are needed.

## Introduction

Congenital heart defects (CHDs) are structural malformations of the heart and/or the great intrathoracic vessels [1]. They are the most frequent congenital malformation [2] and prevalence is estimated to be around 8–10 per 1,000 live births worldwide [2–5]. The most critical and severe defects, the major CHDs, (1–2 per 1,000 live births) have a serious impact on neonatal mortality and morbidity, and frequently result in neonatal heart failure or circulatory collapse requiring acute surgery [5]. However, other types of CHDs, like bicuspid aortic valve and transient septal heart defects, are far more frequent but of limited clinical significance [3, 6].

The aetiology of CHDs is complex and poorly understood [7]. An identifiable underlying cause (genetic and/or environmental) is present in 20–30% of CHDs [6]. Thus, 8–10% of CHDs can be attributed to chromosomal aberrations (e.g., DiGeorge syndrome, Down syndrome, Turner syndrome, etc.) [6, 8]; while 5–15% may be the result of single-nucleotide or pathogenic copy number variants [6, 9]. Additionally, viral infections in pregnancy (e.g., rubella), as well as exposure to certain teratogenic substances (e.g., alcohol or antiepileptic drugs) [7] may cause CHDs if the foetus is exposed at a critical point in development [7].

The normal pregnancy is an adaptive interplay between the maternal metabolism and foetal development [2]. Differentiation of cardiac tissues begins in the third week of gestation, and by week eight, the foetal heart has undergone major changes, and will resemble the postnatal heart in structure and function [1]. Thus, the maternal-foetal interaction in the first two months of pregnancy is likely to be the most relevant window in time for a causative relation between maternal metabolism and CHDs [1]. In first trimester, the foetus does not have the ability to secrete insulin, which may result in foetal hyperglycaemia in the event of relative maternal insulin resistance [10]. Animal models have shown that in embryos of chicks and rodents, exogenous glucose may cause a variety of malformations [11, 12]. The significance of foetal hyperglycaemia has not been demonstrated in humans, but increasingly worse glycaemic control around conception in women with diabetes type 1 (DM1) has recently been reported to be associated with a progressively increased risk of CHDs in the offspring [13].

Large cohort studies from Scandinavia and North America have reported an association between increased risk of CHDs and maternal metabolic disorders such as obesity [14, 15], diabetes [16–18], hypertension [19, 20], and preeclampsia (PE) [21, 22]. All these metabolic disorders can be associated with hyperglycaemia and underlying insulin resistance, and thereby with the metabolic syndrome (MetS) [23, 24]. There is a need to examine the evidence base for associations between combinations of maternal metabolic disorders, e.g., MetS, and CHDs. Either to complement existing knowledge or to define important new areas of research.

MetS includes a cluster of metabolic disorders in combination, usually any three of the following: abdominal obesity, insulin resistance, hypertension, and dyslipidaemia [23, 25]. It is commonly reported to be a risk factor for diabetes type 2 (DM2) and cardiovascular disease in a non-pregnant population [26]. Only a single multi-centre study from Australia, New Zealand and United Kingdom has assessed MetS in pregnant women and reported an incidence of 12.3% at 15 weeks' gestation [27] using the International Diabetes Federation criteria [25]. It is plausible that obesity, diabetes, hypertension, PE and dyslipidaemia as individual conditions, or more likely in combination, during pregnancy are expressions of insulin resistance and MetS [23, 24]. Combinations of several maternal metabolic disorders in the same pregnancy (defined as obesity, diabetes, hypertension, PE, and dyslipidaemia), all related to MetS, are likely to be associated with higher risk of CHDs in the offspring as compared to women with a single disorder. The evidence base for an association between CHDs and combinations of metabolic disorders has not been systematically examined.

The aim of this systematic review is to evaluate the literature of associations between maternal metabolic disorders (obesity, diabetes, hypertension, PE, dyslipidaemia, or MetS) and particularly combinations thereof and CHDs in the offspring. The relation with overall occurrence of CHDs as well as CHDs broken down into individual subtypes. Since these maternal metabolic disorders likely persist over time, they may be relevant for the risk of CHDs in subsequent pregnancies. Furthermore, we examined the extent to which recent reviews have mentioned the possibility of combined maternal disorders. On this background, we will point to gaps in current knowledge and make recommendations for future research possibly paving the way to improved prevention of CHDs.

## Materials and methods

### Search strategy

Following the Systematic Reviews and Meta-Analyses (PRISMA) checklist (S1 Appendix), a systematic literature search of papers published between January 1, 1990 and January 14, 2021 was conducted using PubMed and Embase. A search in a number of clinical trial registers April 12, 2021 (search term: "congenital heart defects") identified five trials that might be relevant for this topic. However, no results were available from the trial registers or PubMed, and therefore not possible to include in the review. Because this review aimed to compare results from high quality, peer-reviewed papers only, the search did not include grey literature. MetS was initially described by G.M. Reaven in 1988 as an "insulin resistance syndrome" [26], and the literature search was limited to publications focussing on its role in CHDs. The search strategy used keywords that combined CHDs and pregnancies with maternal metabolic disorders (obesity, diabetes, hypertension, PE, dyslipidaemia and/or MetS). Although DM1 is not a part of the MetS, it was included in the review as a metabolic disorder because the associated hyperglycaemia is known to be associated with CHDs. More details about the literature search can be found in S1 Table. Subsequently, reference lists from the individual papers were reviewed for additional relevant studies.

### Exposures, outcomes and definitions

The exposures of interest were maternal overweight and obesity, diabetes (DM1, DM2 or gestational diabetes (GDM)), hypertension, PE, dyslipidaemia, and/or MetS during pregnancy. Maternal body mass index (BMI) (weight in kilograms divided by the square of the height in meters ($kg/m^2$)) was defined as pre-pregnancy or early-pregnancy BMI measured in first trimester. The outcome of interest was CHDs defined as structural malformations of the heart chambers, heart valves, great arteries and septal defects, corresponding to DQ20-26

in the World Health Organization International Classification of Diseases (ICD-10) or diagnoses referable to these. The CHD diagnoses are described as "any CHDs" for the whole group of heart defects (only including studies that explicitly listed their definition thereof), "major CHDs" for a group of the most critical and severe CHD diagnoses or as CHD subtypes in relation to the maternal disorders. Studies could include both singleton and multiple births.

## Eligibility criteria

Two different authors (GH, PLH and/or INT) screened all titles and abstracts individually. Studies were initially eligible if they met the following criteria: 1. Studies were published in English; 2. Studies were case-control or cohort studies; 3. The exposures of interest were maternal overweight or obesity, hypertension, PE, diabetes, dyslipidaemia, and/or MetS; and 4. The outcome of interest was CHDs in the offspring. Studies were excluded if they did not relate to the topic; if they did not report CHDs referable to ICD-10 codes (DQ20-26); if they did not have a healthy control group or a non-exposed group; and if the data was presented in an included publication (Fig 1). Furthermore, studies were excluded if they had less than 15 CHD cases in total. In consequence, some studies will only contain few cases of each CHD subtype.

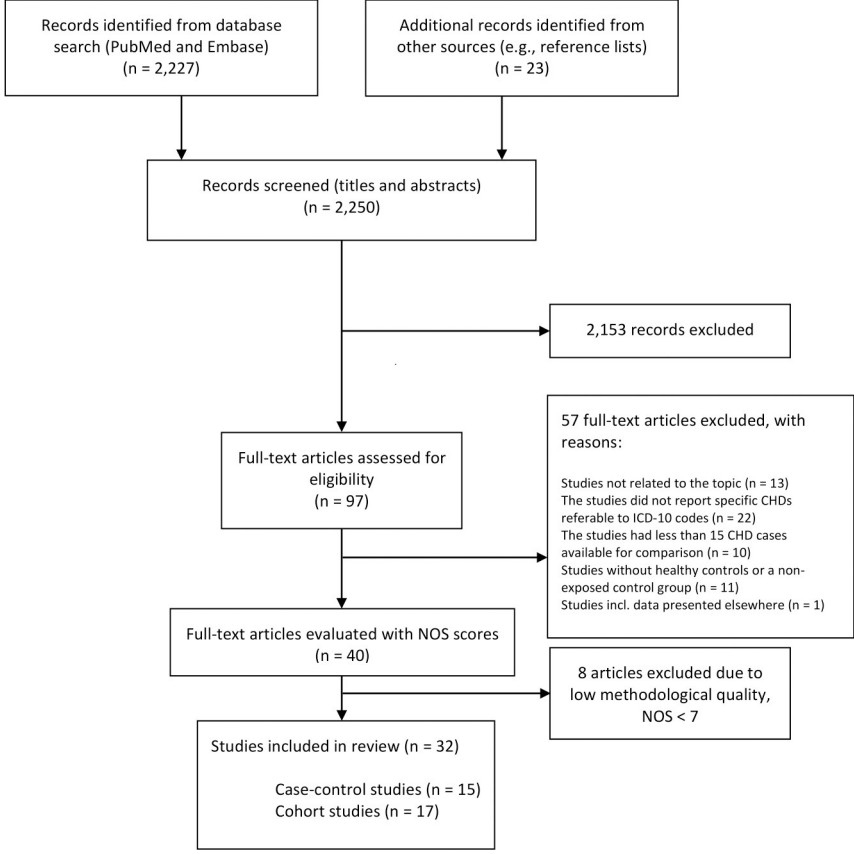

**Fig 1. Flow diagram.** The process of selection of studies that were included in the review. Literature search included studies published from January 1, 1990 to January 14, 2021. Abbreviations: CHDs, congenital heart defects; ICD, International Classification of Disease; NOS, Newcastle-Ottawa Scale.

## Quality assessment

One author (GH) conducted the study selection and quality assessment based on a full-text review. Any doubts were resolved by discussion with at least one co-author. The Newcastle-Ottawa Scale was used to assess the quality of eligible studies [28]. Using this tool (S2 Table), each study was judged on three categories: 1. selection of cohorts or cases and controls; 2. comparability of cohorts or cases and controls on the basis of the design or analysis; and 3. ascertainment of outcome or exposure. Highest quality studies could be awarded nine stars. If a study received seven or more stars (S3 and S4 Tables), it was considered of high methodological quality and included in the review [28].

Data was extracted to figures without processing results. Results are presented as odds ratios (OR), risk ratios (RR) or prevalence ratios (PR) with 95% confidence interval (CI). Results are presented adjusted (aOR/aRR/aPR) for pertinent possible confounding factors, if accessible.

# Results

## Study identification and selection

The selection of studies is described in Fig 1. In the event of multiple publications using the same data, we included the study that provided the most comprehensive information. A total of 32 publications were included in the review (Table 1 and S3 Table). Among these, 17 dealt with maternal overweight or obesity [14, 15, 19, 29–42], 10 dealt with pre-gestational diabetes (PGDM) [16–18, 40, 41, 43–47], four dealt with DM1 [13, 19, 48, 49], two dealt with DM2

**Table 1. Summary table of characteristics of included studies.**

| Characteristics | Range of values |
|---|---|
| Year of publication | 1991–2020 |
| Country of study population | North America (n = 15), Europe (n = 13), Other countries (n = 4) |
| Study design | Cohort (n = 17), Case-control (n = 15) |
| Population size | Cohorts (41 013–4 207 898), Case-controls (525–1 124 370) |
| Cases with CHDs | Cohorts (233–48 249), Case-controls (35–10 625) |
| Included study population | Only live births (n = 15); Live births + stillbirths (n = 9); Live births + stillbirths + terminated pregnancies (n = 8) |
| Plurality | Singletons (n = 13), All (n = 8), Not stated (n = 11) |
| Maternal exposure | Obesity (n = 16), PGDM (n = 10), DM1 (n = 4), DM2 (n = 2), GDM (n = 8), hypertension (n = 4), PE (n = 3), obesity+GDM (n = 1) |
| Sources of maternal exposure | Registers (n = 18), interviews (n = 9), medical records (n = 3), questionnaire + medical records (n = 2) |
| Studies of each CHD subtype | Heterotaxia (n = 9), UVH (n = 6), Conotruncal defects (n = 8), Common truncus (n = 7), TGA (n = 20), ToF (n = 20), DORV (n = 3), Aortic arch defects (n = 1), AVSD (n = 17), TAPVR (n = 8), LVOT (n = 9), HLHS (n = 12), CoA (n = 14), Aortic stenosis (n = 8), RVOT (n = 9), Pulmonary valve stenosis (n = 8), Ebstein's anomaly (n = 5), Septal defects (n = 6), ASD (n = 21), VSD (n = 18) |
| Newcastle-Ottawa Scale (NOS) score | 7–8 |

A more comprehensive version of the table is available as S3 Table. Details on NOS score can be seen in S4 Table. Abbreviations: ASD, atrial septal defects; AVSD, atrioventricular septal defects; CoA, coarctation of the Aorta; DM1, diabetes type 1; DM2, diabetes type 2; DORV, double outlet right ventricle; GDM, gestational diabetes; HLHS, hypoplastic left heart syndrome; LVOT, left ventricular outflow tract defects; PE, preeclampsia; PGDM, pre-gestational diabetes; RVOT, right ventricular outflow tract defects; TAPVR, total anomalous pulmonary venous return; TGA, transposition of the great arteries; ToF, Tetralogy of Fallot; UVH, univentricular heart; VSD ventricular septal defects.

[19, 48], eight dealt with GDM [16–18, 39–41, 46, 49], four dealt with hypertension [19, 20, 43, 48], three dealt with PE [21, 22, 50], and none were about dyslipidaemia or MetS (as a diagnostic category). Except for the combination of PGDM and GDM, six studies investigated more than one maternal metabolic disorder (but not in combination) [19, 39–41, 43, 48], and one study assessed a combination of two conditions (obesity and GDM) and the risk of CHDs [31].

## Study characteristics

The 32 included studies consisted of 17 cohort studies and 15 case-control studies. Characteristics of the 32 studies are presented in Table 1 and S3 Table. Three examined populations of non-European descent (Taiwan and China) [39, 42, 48]. Half of the studies reported results from livebirths only; eight studies had populations including both singleton and multiple births; and 11 studies did not state whether they included only singleton or multiple births. Associations between maternal metabolic disorders and CHDs are illustrated in Fig 2 (any CHDs) and S1–S20 Figs (CHD subtypes). All 32 studies were assessed using the Newcastle-Ottawa Scale and were required to be of high methodological quality (S2 and S4 Tables). Study

**Fig 2. Maternal metabolic disorders and risk of any congenital heart defect in the offspring.** Any CHDs defined as the whole group of heart defects (only including studies that explicitly listed their definition thereof). Obesity is defined as BMI $\geq$ 30 kg/m$^2$ unless other is stated; early-onset PE defined as diagnosed before gestational week 34; and PGDM are defined as DM1 or DM2. All risk estimates are adjusted unless other is stated; *, not adjusted; §, isolated defects; ¤, obesity defined from ICD-10 codes; £, BMI $\geq$ 35 kg/m$^2$; ¥, BMI $\geq$ 28 kg/m$^2$; †, BMI 35-<40 kg/m$^2$; @, BMI 30.0–39.9; #, untreated hypertension; ‡, estimate only for severe CHDs. Abbreviations: BMI, body mass index; DM1, diabetes type 1; DM2, diabetes type 2; GDM, gestational diabetes; OR, odds ratio; PE, preeclampsia; PGDM, pre-gestational diabetes; PR, prevalence ratio; RR, risk ratio.

designs were heterogeneous, particularly regarding the definition and categorization of the maternal metabolic disorders.

## Maternal overweight or obesity and CHDs in the offspring

Several studies have investigated the association between maternal overweight or obesity and any CHDs [14, 15, 19, 31, 33, 37, 39, 41, 42]. In Fig 2, five out of eight studies found a significant association between maternal obesity (BMI $\geq$ 30 kg/m$^2$) and any CHDs. Four of them reported higher risk estimates for any CHDs with higher maternal BMI: RR 1.07 (BMI 25–29.9 kg/m$^2$) and RR 1.60 (BMI $\geq$ 40 kg/m$^2$) [14]; aOR 1.15 (BMI 25–29.9 kg/m$^2$) and aOR 1.34 (BMI $\geq$ 40 kg/m$^2$) [15]; aOR 1.16 (BMI 25–29.9 kg/m$^2$) and aOR 1.31 (BMI $\geq$ 35 kg/m$^2$) [31]; and aOR 1.00 (BMI 25–29.9 kg/m$^2$) and aOR 1.33 (BMI $\geq$ 40 kg/m$^2$) [32]. The fifth study only reported an association between maternal obesity diagnosed as ICD-10 codes and any CHDs, and did not assess the risk in relation to BMI categories [19]. Three studies found no significant association between any CHDs in the offspring and maternal overweight or obesity [37, 39, 41]. Yuan et al. only had 11 pregnant women with BMI > 28 kg/m$^2$, which were grouped in an overweight category (BMI > 24 kg/m$^2$) and therefore not comparable with the other results [42].

Most studies investigated specific CHD subtypes in relation to maternal BMI, these results are presented for maternal obesity (BMI $\geq$ 30 kg/m$^2$) in S1–S20 Figs and summarized in Table 2. Some studies found significant associations between obesity and 13 CHD subtypes; however, other studies did not find a significant association.

**Table 2. Overview of association between congenital heart defect subtypes and maternal exposures.**

| | Obesity | PGDM | DM1 | DM2 | GDM | Hypertension | PE (early-onset) | Obesity + GDM |
|---|---|---|---|---|---|---|---|---|
| Heterotaxia | 2↓ | 3(2)↑ | 1↑ | 1(1)↑ | 2↑ | 1↑ 1↓ | 2(1)↑ 1↓ | 1↑ |
| UVH | 2↑ | 2(1)↑ | NA | NA | 1↑ | 1↑ 1↓ | 1(1)↑ | 1↓ |
| Conotruncal defects | 4(2)↑ 1↓ | 3(2)↑ | 1(1)↑ | 1(1)↑ | 1↑ | 2(1)↑ | 1(1)↑ | 1↑ |
| Common truncus | 1↑ | 3(2)↑ | NA | NA | 1(1)↑ | 2(1)↑ | 1(1)↑ | NA |
| TGA | 8(2)↑ 3↓ | 6(3)↑ | NA | NA | 2↑ | 2↓ | 2↓ | 1↓ |
| ToF | 7(3)↑ 4↓ | 5(4)↑ | NA | NA | 2(1)↑ | 2(1)↑ | 1↑ 1↓ | 1(1)↑ |
| DORV | 1↑ | 1(1)↑ | NA | NA | NA | NA | 1(1)↑ | NA |
| Aortic arch defects | 1(1)↑ | NA | NA | NA | NA | NA | NA | NA |
| AVSD | 6(2)↑ | 6(5)↑ | 1(1)↑ | 1(1)↑ | 3(1)↑ 1↓ | 3(2)↑ | 3(3)↑ | 1↑ |
| TAPVR | 1↑ 1↓ | 4(2)↑ | NA | NA | 1(1)↑ 1↓ | 1↓ | 1(1)↑ | 1↑ |
| LVOT | 2(1)↑ 2↓ | 4(4)↑ | 1(1)↑ | 1(1)↑ | 2(1)↑ | 1↑ | 1↓ | 1(1)↑ |
| HLHS | 5(2)↑ 1↓ | 3(1)↑ | NA | NA | 1↑ 1↓ | 1↑ 1↓ | 1↑ 1↓ | 1(1)↑ |
| CoA | 5↑ 1↓ | 5(3)↑ | NA | NA | 2(1)↑ | 1(1)↑ 1↓ | 2↑ | 1↑ |
| Aortic stenosis | 2(1)↑ 1↓ | 4(3)↑ | NA | NA | 1(1)↑ 1↓ | 1↑ 1↓ | NA | 1↑ |
| RVOT | 3(2)↑ 1↓ | 4(3)↑ | 1↑ | 1↑ | 2(1)↑ | 1(1)↑ | 1(1)↑ | 1(1)↑ |
| Pulmonary valve stenosis | 4(2)↑ | 3(2)↑ | NA | NA | 2(1)↑ | 1(1)↑ | NA | 1(1)↑ |
| Ebstein's anomaly | 1↑ | 1(1)↑ | NA | NA | 1↑ | 2↑ | 1↑ | NA |
| Septal defects | 3(1)↑ | 2(2)↑ | NA | NA | 1(1)↑ | NA | 1(1)↑ | 1(1)↑ |
| VSD | 10(3)↑ | 7(5)↑ | 1(1)↑ | 1(1)↑ | 2(1)↑ | 3(3)↑ | 3(3)↑ | 1↑ |
| ASD | 9(6)↑ | 5(3)↑ 1↓ | 1(1)↑ | 1(1)↑ | 2(1)↑ | 2(2)↑ 1↓ | 2(2)↑ | 1(1)↑ |

Numbers reflect number of studies examining specific combinations of CHD subtype and exposure. Numbers of significant associations are given in bold and brackets. Arrows reflect positive (↑) and negative (↓) directions of the associations. The individual studies are detailed in S1–S20 Figs. Abbreviations: ASD, atrial septal defects; AVSD, atrioventricular septal defects; CoA, coarctation of the Aorta; DM1, diabetes type 1; DM2, diabetes type 2; DORV, double outlet right ventricle; GDM, gestational diabetes; HLHS, hypoplastic left heart syndrome; LVOT, left ventricular outflow tract defects; PE, preeclampsia; PGDM, pre-gestational diabetes; RVOT, right ventricular outflow tract defects; TAPVR, total anomalous pulmonary venous return; TGA, transposition of the great arteries; ToF, Tetralogy of Fallot; UVH, univentricular heart; VSD ventricular septal defects.

### Maternal pre-gestational diabetes and CHDs in the offspring

Most studies defined PGDM as either DM1 or DM2 (S3 Table). Only four studies assessed DM1 or DM2 individually [13, 19, 48, 49]. Seven studies found a strong significant association with any CHDs (Fig 2), irrespective of the type of diabetes investigated, PGDM [16–18, 46], DM1 [13, 19, 48], or DM2 [19, 48]. One study by Dolk et al. had four exposed CHD cases and presented a trend (Fig 2) [41].

Two studies presented associations between PGDM and major CHDs, but they defined major CHDs differently: Leirgul et al. found an association between PGDM and major CHDs (aRR 3.34, CI 2.48–4.49) [18], and Chou et al. found an association between DM2 and major CHDs (aOR 2.80, CI 2.04–3.85), but did not find a significant association between DM1 and major CHDs (aOR 0.73, 0.10–5.32) [48].

Several studies have investigated the relationship between PGDM, DM1 or DM2 and specific CHD diagnoses [16–19, 40, 43–47, 49] and in many cases found a strong significant association (S1–S20 Figs and Table 2).

### Maternal gestational diabetes and CHDs in the offspring

Five out of six studies reported a significant association between GDM and any CHDs [16–18, 39, 41, 46], but only the results from five of them are shown in Fig 2. The last study by Øyen et al. found an association between any CHDs and GDM diagnosed in the third trimester (aRR 1.36; CI 1.07–1.69), but did not find a significant association between any CHDs and GDM diagnosed in the second trimester [16]. Five studies reported a weak association between GDM and some subtypes of CHDs (S1–S20 Figs and Table 2).

### Maternal chronic hypertension and CHDs in the offspring

Two large cohort studies from Canada and Taiwan reported a significant association between maternal pre-existing chronic hypertension and any CHDs (Fig 2) [19, 48]. Two studies assessed the association between subtypes of CHDs and chronic hypertension [19, 43], (S1–S20 Figs), but associations were inconsistent. None of the three studies mentioned reported on potential antihypertensive drug use by the mothers. An American case-control study with 10,625 cases with CHDs found a significant association between untreated hypertension and any CHDs (Fig 2) as well as for five subtypes of CHDs (S1–S20 Figs and Table 2) [20].

### Maternal preeclampsia or gestational hypertension and CHDs in the offspring

Three studies investigated associations between PE and any CHDs or major CHDs. Although the definition of PE varied among the studies, all studies defined early-onset PE as diagnosed before gestational week 34, and the association with any CHDs can be seen in Fig 2. Pregnancies with PE were associated with any CHDs (PR 1.57, CI 1.48–1.67) [22], and early-onset PE was associated with major CHDs (aRR 2.7, CI 1.3–5.6 [50], and aPR 3.64, CI 2.17–6.10 [22]), all compared to normotensive pregnancies.

Weaker associations, between PE and any CHDs, were reported when PE was diagnosed later in pregnancy: PE with delivery at 34 to 36 weeks (OR 2.82, CI 2.38–3.34) [21]; PE at term (OR 1.16, CI 1.06–1.27) [21]; and late-onset PE (> 34 weeks' gestation) (aPR 1.14, CI 1.06–1.23) [22]. Furthermore, Boyd et al. did not find an association between gestational hypertension and any CHDs [21]. The studies also assessed subtypes of CHDs and their association with early-onset PE, which are illustrated in S1–S20 Figs and summarized in Table 2, especially atrioventricular septal defects (S9 Fig) and atrial septal defects (S19 Fig) were strongly associated with early-onset PE.

### Risk of CHDs in the offspring in subsequent pregnancies

Two studies assessed PE and CHDs, and potential risks of both in subsequent pregnancies [21, 50]. While, Brodwall et al. did not find an association between PE and CHDs across pregnancies [50], Boyd et al. reported a much higher risk of having a child with CHDs in subsequent pregnancies after a pregnancy with PE and delivery before 34 weeks' gestation (OR 7.91, CI 6.06–10.3). Boyd et al. also found an association between a previous pregnancy with a child with CHDs and increased risk of PE with delivery before 34 weeks' gestation in subsequent pregnancies (OR 2.37; CI 1.68–3.34) [21]. Studies on the other maternal metabolic disorders did not comprise an analysis of risks in subsequent pregnancies.

### Combination of maternal obesity and gestational diabetes and CHDs in the offspring

Only one study comprised an assessment of the risk associated with pregnancies affected by two maternal metabolic conditions. Gilboa et al. reported a possible additive effect of the combination of maternal obesity (BMI $\geq$ 30 kg/m$^2$) and GDM and the risk of any CHDs compared to only obesity (aOR 1.82 vs. 1.31) [31]. The relation between the combination of maternal obesity and GDM and subtypes of CHDs are reported in S1–S20 Figs and summarized in Table 2 showing a significant association for six subtypes of CHDs.

### Trends in associations of specific CHD subtypes

As shown in Table 2, there are no clear differences in the direction of associations between CHD subtypes and individual maternal metabolic disorders. Of note is that none of the studies, suggesting reduced risk of CHD was statistically significant.

### Recent reviews

Systematic reviews and meta-analyses further documented the associations for some of the individual maternal metabolic disorders: obesity [51–53], diabetes [17, 54, 55], and hypertension [56]. However, only a single review mentioned combinations of GDM and obesity, citing only two studies [57]. No reviews sought to estimate the effect of simultaneous presence of several metabolic disorders.

## Discussion

### Main findings

The associations between any CHDs and maternal metabolic disorders fall in three groups: A. chronic hypertension, GDM, and obesity with OR/RR/PR of 1.2–1.9; B. PGDM with OR/RR/PR around 3–4; and C. early-onset PE with OR/RR/PR of 5–7. Thus, for the most prevalent maternal conditions, the associated risks of CHDs are only marginally elevated. Although maternal obesity is associated with higher risk of GDM, DM2, hypertension, PE, and dyslipidaemia [58], no studies assessing the association between CHDs in the offspring and the combination of all maternal metabolic disorders were identified, and only one study dealt with the combination of two maternal metabolic conditions (obesity and GDM) as risk factor for CHDs in the offspring [31]. All other studies investigated the individual metabolic disorders and their association with CHDs, and there were no eligible studies on dyslipidaemia. Overall, those studies found a significant association between any CHDs and obesity, PGDM, GDM, hypertension, or early-onset PE, respectively (Fig 2). Contradictory evidence of associations with subtypes of CHDs were noted for most maternal conditions, with the exception of

early-onset PE, which was strongly associated with both atrioventricular septal defects (S9 Fig) and atrial septal defects (S19 Fig); and PGDM which was strongly associated with most sub-types (S1–S20 Figs).

Early-onset PE was strongly associated with a higher risk of having a child with CHD in subsequent pregnancies, studies on the other maternal metabolic disorders did not approach risks in subsequent pregnancies.

There was no discernible pattern of associations between CHD subtypes and metabolic disorders (Table 2). However, the number of studies for many of the combinations is very small.

## Aetiological considerations and possible prevention of CHDs

**Relation between maternal metabolic disorders and foetal heart development.**   There is a clear pathophysiological connection between maternal insulin resistance and foetal CHDs, with glucose-mediated mechanisms of CHD involving multiple developmental pathways [59]: comprise left-right patterning [60], alterations in neural crest cells migration and formation [61, 62], increased apoptosis [63, 64] as well as changes in nitric oxide signalling [65], and impaired autophagy [66]. All these mechanisms may cause or modify the development of CHDs. The investigated maternal metabolic disorders (obesity, diabetes, hypertension, pre-eclampsia, dyslipidaemia, and MetS) are all linked to insulin resistance and/or hyperglycaemia [24, 67]. The finding of the same association for these conditions as for DM1 [16] justifies a particular focus on the role of aberrations in the glucose-insulin homeostasis. Obesity, DM1, DM2, chronic hypertension, and dyslipidaemia are present prior to conception and therefore affect maternal metabolism and foetal development during the essential first two months of pregnancy. However, PE and GDM are per definition not diagnosed before mid-pregnancy [68, 69], but it is likely that subclinical metabolic alterations emerge at a much earlier stage. It can be challenging to distinguish GDM from PGDM since most women are not screened for diabetes before pregnancy, and up to 50% of women diagnosed with GDM develop DM2 within five years after giving birth [70]. So, it is plausible that many women with GDM might already be insulin resistant without it being detectable in first trimester. This is particularly challenging because fasting and postprandial glucose concentrations are normally lower in the early part of pregnancy (e.g., first trimester and first half of second trimester) than in normal, non–pregnant women [69]. Insulin resistance is more pronounced early in pregnancies with PE compared to pregnancies without PE [24, 71]. In first trimester pregnancies that later develop PE are characterised by changes in IGF-axis markers, e.g. PAPP-A, and ProMBP [72, 73], placental proteins [74], and adipokines, e.g. leptin and adiponectin [75, 76], indicating that the clinical presentation of PE is preceded by a foeto-maternal dyscrine condition. The fact that the nutritional status of the foetus influences the risk of cardiovascular disease in adulthood (Barker Hypothesis) [77]—also indicates that placental function may play a role in the aetiology of foetal malformations. There is a need to identify particular pathways where perturbations, especially in the glucose-insulin axis, cause specific malformations, but such studies will have to be large as some of the malformations are fairly rare and the molecular aetiology likely heterogeneous.

**Possible teratogenic effect of medication.**   Women with chronic hypertension or DM1 will most likely be undergoing medical treatment as well as some women with DM2 will be treated with antidiabetic drugs. Some of these drugs might have a teratogenic effect and by not taking them into considerations in the studies, they could influence the estimates of association. It is thus important to distinguish between the effects of the drugs and the conditions proper. Insulin is not believed to be teratogenic [78], suggesting that the focus should be on the insulin resistance and the glucose levels. Observational studies suggest that exposure to

either metformin or beta-blockers during pregnancy may actually increase the risk of certain types of CHDs in the foetus [20, 79]. A meta-analysis by Ramakrishnan et al. observed an association between untreated hypertension and CHDs, which suggests that the association between hypertension and CHDs is not simply due to teratogenic effects of medication alone, but the effect was larger for treated hypertension [56]. Thus, antihypertensive medications may lead to an additional increase in risk, but it might also just indicate a more severe disease in the mother.

**Prevention of CHDs in the offspring.** It is very well-established that maternal obesity is associated with marginally increased risk of CHDs, and two meta-analyses reported a dose-response effect on risk of CHDs with increasing BMI [53, 80]. But an increase of 20–40% of a very small risk in general may not be clinically significant. Particularly as it has not yet been shown that weight reduction normalizes the risk and a study showing such an effect would be complicated and difficult to perform. Future studies investigating the prevalence of CHDs in obese pregnant women could compare women who had undergone bariatric surgery with a control group as well as risk in pregnancies before and after the surgery as it has been done in studies on birth weight and obesity in childhood [81, 82]. The Swedish cohort study of 1.2 million pregnant women with DM1 showed that poor glycaemic control around conception was associated with a progressively higher risk of CHDs in the offspring [13]. PGDM and early-onset PE have a strong association with CHDs, however, the clinical significance of preventive measures has not been established.

## Heterogeneity in studies

Comparison of results from different studies in this review was hampered by heterogeneity. This is due to several factors: 1. Case-control studies provide estimates of risk that may be less valid and reliable than prospective, population-based cohort studies for estimating occurrence of CHDs–i.e., for some subtypes of CHDs, the cohort studies find a significant association while the results from the case-control studies are insignificant. 2. The studies assessing maternal obesity and CHDs often excluded pregnancies with PGDM, whereas others did not and are therefore difficult to compare; studies on obesity usually also divided maternal BMI into categories that distinguished them from a direct comparison with another study. 3. Diagnostic criteria for PGDM and GDM were not always explained. 4. Not all studies include stillbirths and terminated pregnancies, which could skew the results, i.e., the risk of CHDs in live born offspring of women with obesity relative to women without obesity might be overestimated, since it is well-known that congenital malformations are more difficult to detect by prenatal ultrasound in these women and they are therefore not necessarily given the possibility to terminate the pregnancy in case of a severe malformation [83]. 5. Studies on both singleton and multiple births without distinguishing between them may add to the heterogeneity because the risk of CHDs is higher in multiple birth [84].

We presented adjusted risks and compared them although the studies did not all adjust for the same confounders. Many studies adjust for maternal age, parity and year of birth, while others also adjust for race/ethnicity, education, smoking, marital status etc. Even though not always available, we could have presented crude risks, however, we believe that the adjusted risks represent the best possible estimate since results otherwise would have been known to be biased.

## Significance of genetic aetiology

The aetiology of CHD is complex and the important breakthroughs in next generation sequencing of whole genomes as well as implementation of chromosomal micro arrays have as of now led to the identification of a strong genetic component in 15–20% of CHD cases [6].

The genetic aetiology ranges from chromosomal disorders over micro-deletions to copy number variants and single-gene disorders. Many of the genetic causes of CHDs are characterized by extensive phenotypic variability and a broad spectrum of comorbidities [6]. Thus, some cases of e.g. Williams syndrome [85] as well as Bardet-Biedl syndrome [86] exhibit malformations in combinations with obesity, DM2, and hypertension. In cases where the mother carries genetic variants that cause or pre-dispose to such conditions, the association between CHDs and maternal metabolic disorders might be one of common aetiology (horizontal pleiotropy) rather than one of a causal relation between maternal conditions and the development of CHDs (vertical pleiotropy). With further advances in genetic characterization of CHDs, one might see a more detailed understanding of the extent of comorbidities in CHD-associated genetic disorders. This phenomenon might bias the association analysis.

## Strengths and limitations

The strength of this review is that we have selected only high-quality, original research studies from a systematic literature search. However, we have limited our selection of studies to those published in English and from 1990 and after, which may have excluded a number of high-quality studies, which were published in another language or before 1990. However, the screening of the references of the articles included did not reveal relevant studies published before 1990. Furthermore, our review did not include meta-analyses as prescribed by PRISMA since we found the data too heterogeneous to summarize in a meta-analysis. Finally, it should be remembered that some CHD subtypes are very rare, resulting in very broad confidence intervals in association analyses.

## Perspectives

The global birth prevalence of CHDs has increased by 10% every five years between 1970 and 2017, which probably is due to increased detection of milder lesions [3]. The incidence of MetS often parallels the incidence of obesity, which has nearly tripled worldwide since 1975 [87], and Centers for Disease Control and Prevention estimated that 40% of women aged 20–39 years old in the United States had a BMI $\geq$ 30 kg/m$^2$ in 2017–2018 [88]. Maternal obesity and dyslipidaemia increases the risk of developing both PE and GDM [89, 90], GDM closely associated with risk of DM2, PE increases the risk of future chronic hypertension [91], and studies indicate that women with PE have increased risk of insulin resistance later in life [92]. So, the maternal metabolic disorders might be related to later MetS (except for DM1), and all components are associated with CHDs in offspring–either individually or with an additive effect. Our analysis shows that there is a remarkable lack of knowledge on several aspects of the link between maternal metabolic disorders and CHDs in the offspring. Large population-based studies are needed to identify possible synergistic effects. Indeed, the combination may have a greater impact than each individual factor and result in amplified risk of CHDs–also taking into consideration at what time the maternal metabolic disorder develops in relationship to the development of the foetal heart. Furthermore, such studies should be ethnically inclusive, and also include women of different socio-economic status. There is also a need for more detailed studies of the molecular mechanisms of CHDs, especially regarding the possibly common underlying pathophysiological mechanisms, insulin resistance and glucose-insulin homeostasis in particular, and their genomic basis. A combination of such studies in carefully investigated cohorts and the use of electronic health records in a population-wide biobank setting could bring relevant knowledge on mechanisms that might pave the way to preventive approaches to CHDs. Furthermore, it is important to examine whether a modification of a pre-existing condition influences risks for CHDs.

## Conclusion

CHDs are the most common malformations and a major cause of morbidity and mortality in childhood. Although the aetiology is complex, it is important to identify risk factors in order to pave the way for targeted prevention of CHDs. Our review shows that there is a well-established association, albeit to a varying extent, between individual maternal metabolic disorders such as obesity, diabetes, hypertension, and PE with increased risk of CHDs in the child. However, the evidence base for association between combinations of maternal metabolic disorders and CHDs was found to be very small. Since occurrence of MetS is rapidly increasing, there is a need for such studies. Secondly, there is a need for large studies based on genetically characterized CHD cases, where the association with MetS could be an expression of comorbidity involving both CHDs and maternal metabolic disorders implying aberrations of the glucose-insulin homeostasis. Thirdly, there is a need for studies covering populations other than those of European and Asian descent, particularly as the prevalence of MetS and its different manifestations vary widely around the globe. Such studies may result in the identification of subgroups of women, in whom MetS might be of particularly importance as a risk factor. Finally, it is important to examine whether preventive actions against pre-pregnancy maternal metabolic disorders influences the risk of CHDs.

## Supporting information

**S1 Fig. Maternal metabolic disorders and risk of heterotaxia in the offspring.** Obesity is defined as BMI $\geq$ 30 kg/m$^2$ unless other is stated; early-onset PE defined as debut before gestational week 34; PGDM are defined as DM1 or DM2; all risk estimates are adjusted unless other is stated; *, not adjusted; §, isolated defects; ¤, obesity defined from ICD-10 codes; £, BMI $\geq$ 35 kg/m$^2$; €, BMI > 29 kg/m$^2$; †, BMI 35-<40 kg/m$^2$; #, untreated hypertension; $, Brodwall et al. pooled early-onset PE and severe PE. Abbreviations: BMI, body mass index; DM1, diabetes type 1; DM2, diabetes type 2; GDM, gestational diabetes; OR, odds ratio; PE, preeclampsia; PGDM, pregestational diabetes; PR, prevalence ratio; RR, risk ratio.
(TIF)

**S2 Fig. Maternal metabolic disorders and risk of univentricular heart in the offspring.** Obesity is defined as BMI $\geq$ 30 kg/m$^2$ unless other is stated; early-onset PE defined as debut before gestational week 34; PGDM are defined as DM1 or DM2; all risk estimates are adjusted unless other is stated; *, not adjusted; £, BMI $\geq$ 35 kg/m$^2$; †, BMI 35-<40 kg/m$^2$; #, untreated hypertension. Abbreviations: BMI, body mass index; DM1, diabetes type 1; DM2, diabetes type 2; GDM, gestational diabetes; OR, odds ratio; PE, preeclampsia; PGDM, pregestational diabetes; PR, prevalence ratio; RR, risk ratio.
(TIF)

**S3 Fig. Maternal metabolic disorders and risk of conotruncal defects in the offspring.** Obesity is defined as BMI $\geq$ 30 kg/m$^2$ unless other is stated; early-onset PE defined as debut before gestational week 34; PGDM are defined as DM1 or DM2; all risk estimates are adjusted unless other is stated; *, not adjusted; §, isolated defects; ¤, obesity defined from ICD-10 codes; £, BMI $\geq$ 35 kg/m$^2$. Abbreviations: BMI, body mass index; DM1, diabetes type 1; DM2, diabetes type 2; GDM, gestational diabetes; OR, odds ratio; PE, preeclampsia; PGDM, pregestational diabetes; PR, prevalence ratio; RR, risk ratio.
(TIF)

**S4 Fig. Maternal metabolic disorders and risk of common truncus in the offspring.** Obesity is defined as BMI $\geq$ 30 kg/m$^2$ unless other is stated; early-onset PE defined as debut before

gestational week 34; PGDM are defined as DM1 or DM2; all risk estimates are adjusted unless other is stated; *, not adjusted; #, untreated hypertension. Abbreviations: BMI, body mass index; DM1, diabetes type 1; DM2, diabetes type 2; GDM, gestational diabetes; OR, odds ratio; PE, preeclampsia; PGDM, pregestational diabetes; PR, prevalence ratio; RR, risk ratio.
(TIF)

**S5 Fig. Maternal metabolic disorders and risk of transposition of the great arteries in the offspring.** Obesity is defined as BMI $\geq$ 30 kg/m$^2$ unless other is stated; early-onset PE defined as debut before gestational week 34; PGDM are defined as DM1 or DM2; all risk estimates are adjusted unless other is stated; *, not adjusted; §, isolated defects; £, BMI $\geq$ 35 kg/m$^2$; €, BMI > 29 kg/m$^2$; †, BMI 35-<40 kg/m$^2$; #, untreated hypertension; $, Brodwall et al. pooled early-onset PE and severe PE. Abbreviations: BMI, body mass index; DM1, diabetes type 1; DM2, diabetes type 2; GDM, gestational diabetes; OR, odds ratio; PE, preeclampsia; PGDM, pregestational diabetes; PR, prevalence ratio; RR, risk ratio.
(TIF)

**S6 Fig. Maternal metabolic disorders and risk of Tetralogy of Fallot in the offspring.** Obesity is defined as BMI $\geq$ 30 kg/m$^2$ unless other is stated; early-onset PE defined as debut before gestational week 34; PGDM are defined as DM1 or DM2; all risk estimates are adjusted unless other is stated; *, not adjusted; §, isolated defects; £, BMI $\geq$ 35 kg/m$^2$; €, BMI > 29 kg/m$^2$; †, BMI 35-<40 kg/m$^2$; #, untreated hypertension; $, Brodwall et al. pooled early-onset PE and severe PE. Abbreviations: BMI, body mass index; DM1, diabetes type 1; DM2, diabetes type 2; GDM, gestational diabetes; OR, odds ratio; PE, preeclampsia; PGDM, pregestational diabetes; PR, prevalence ratio; RR, risk ratio.
(TIF)

**S7 Fig. Maternal metabolic disorders and risk of double outlet right ventricle in the offspring.** Obesity is defined as BMI $\geq$ 30 kg/m$^2$ unless other is stated; early-onset PE defined as debut before gestational week 34; PGDM are defined as DM1 or DM2; all risk estimates are adjusted unless other is stated; *, not adjusted. Abbreviations: BMI, body mass index; DM1, diabetes type 1; DM2, diabetes type 2; GDM, gestational diabetes; OR, odds ratio; PE, preeclampsia; PGDM, pregestational diabetes; PR, prevalence ratio; RR, risk ratio.
(TIF)

**S8 Fig. Maternal metabolic disorders and risk of aortic arch defects in the offspring.** Obesity is defined as BMI $\geq$ 30 kg/m$^2$ unless other is stated; early-onset PE defined as debut before gestational week 34; PGDM are defined as DM1 or DM2; all risk estimates are adjusted unless other is stated; *, not adjusted; †, BMI 35-<40 kg/m$^2$. Abbreviations: BMI, body mass index; DM1, diabetes type 1; DM2, diabetes type 2; GDM, gestational diabetes; OR, odds ratio; PE, preeclampsia; PGDM, pregestational diabetes; PR, prevalence ratio; RR, risk ratio.
(TIF)

**S9 Fig. Maternal metabolic disorders and risk of atrioventricular septal defects in the offspring.** Obesity is defined as BMI $\geq$ 30 kg/m$^2$ unless other is stated; early-onset PE defined as debut before gestational week 34; PGDM are defined as DM1 or DM2; all risk estimates are adjusted unless other is stated; *, not adjusted; §, isolated defects; ¤, obesity defined from ICD-10 codes; £, BMI $\geq$ 35 kg/m$^2$; €, BMI > 29 kg/m$^2$; #, untreated hypertension; $, Brodwall et al. pooled early-onset PE and severe PE. Abbreviations: BMI, body mass index; DM1, diabetes type 1; DM2, diabetes type 2; GDM, gestational diabetes; OR, odds ratio; PE, preeclampsia; PGDM, pregestational diabetes; PR, prevalence ratio; RR, risk ratio.
(TIF)

**S10 Fig. Maternal metabolic disorders and risk of total anomalous pulmonary venous return in the offspring.** Obesity is defined as BMI $\geq$ 30 kg/m$^2$ unless other is stated; early-onset PE defined as debut before gestational week 34; PGDM are defined as DM1 or DM2; all risk estimates are adjusted unless other is stated; $^*$, not adjusted; §, isolated defects; £, BMI $\geq$ 35 kg/m$^2$; $\Delta$, estimates for anomalous pulmonary venous return; #, untreated hypertension. Abbreviations: BMI, body mass index; DM1, diabetes type 1; DM2, diabetes type 2; GDM, gestational diabetes; OR, odds ratio; PE, preeclampsia; PGDM, pregestational diabetes; PR, prevalence ratio; RR, risk ratio.
(TIF)

**S11 Fig. Maternal metabolic disorders and risk of left ventricular outflow tract obstruction in the offspring.** Obesity is defined as BMI $\geq$ 30 kg/m$^2$ unless other is stated; early-onset PE defined as debut before gestational week 34; PGDM are defined as DM1 or DM2; all risk estimates are adjusted unless other is stated; $^*$, not adjusted; §, isolated defects; ¤, obesity defined from ICD-10 codes; £, BMI $\geq$ 35 kg/m$^2$. Abbreviations: BMI, body mass index; DM1, diabetes type 1; DM2, diabetes type 2; GDM, gestational diabetes; OR, odds ratio; PE, preeclampsia; PGDM, pregestational diabetes; PR, prevalence ratio; RR, risk ratio.
(TIF)

**S12 Fig. Maternal metabolic disorders and risk of hypoplastic left heart syndrome in the offspring.** Obesity is defined as BMI $\geq$ 30 kg/m$^2$ unless other is stated; early-onset PE defined as debut before gestational week 34; PGDM are defined as DM1 or DM2; all risk estimates are adjusted unless other is stated; $^*$, not adjusted; §, isolated defects; £, BMI $\geq$ 35 kg/m$^2$; €, BMI > 29 kg/m$^2$; #, untreated hypertension; \$, Brodwall et al. pooled early-onset PE and severe PE. Abbreviations: BMI, body mass index; DM1, diabetes type 1; DM2, diabetes type 2; GDM, gestational diabetes; OR, odds ratio; PE, preeclampsia; PGDM, pregestational diabetes; PR, prevalence ratio; RR, risk ratio.
(TIF)

**S13 Fig. Maternal metabolic disorders and risk of coarctation of the aorta in the offspring.** Obesity is defined as BMI $\geq$ 30 kg/m$^2$ unless other is stated; early-onset PE defined as debut before gestational week 34; PGDM are defined as DM1 or DM2; all risk estimates are adjusted unless other is stated; $^*$, not adjusted; §, isolated defects; £, BMI $\geq$ 35 kg/m$^2$; €, BMI > 29 kg/m$^2$; #, untreated hypertension; \$, Brodwall et al. pooled early-onset PE and severe PE. Abbreviations: BMI, body mass index; DM1, diabetes type 1; DM2, diabetes type 2; GDM, gestational diabetes; OR, odds ratio; PE, preeclampsia; PGDM, pregestational diabetes; PR, prevalence ratio; RR, risk ratio.
(TIF)

**S14 Fig. Maternal metabolic disorders and risk of aortic stenosis in the offspring.** Obesity is defined as BMI $\geq$ 30 kg/m$^2$ unless other is stated; early-onset PE defined as debut before gestational week 34; PGDM are defined as DM1 or DM2; all risk estimates are adjusted unless other is stated; $^*$, not adjusted; §, isolated defects; £, BMI $\geq$ 35 kg/m$^2$; #, untreated hypertension. Abbreviations: BMI, body mass index; DM1, diabetes type 1; DM2, diabetes type 2; GDM, gestational diabetes; OR, odds ratio; PE, preeclampsia; PGDM, pregestational diabetes; PR, prevalence ratio; RR, risk ratio.
(TIF)

**S15 Fig. Maternal metabolic disorders and risk of right ventricular outflow tract obstruction in the offspring.** Obesity is defined as BMI $\geq$ 30 kg/m$^2$ unless other is stated; early-onset PE defined as debut before gestational week 34; PGDM are defined as DM1 or DM2; all risk

estimates are adjusted unless other is stated; *, not adjusted; §, isolated defects; £, BMI ≥ 35 kg/m², #, untreated hypertension. Abbreviations: BMI, body mass index; DM1, diabetes type 1; DM2, diabetes type 2; GDM, gestational diabetes; OR, odds ratio; PE, preeclampsia; PGDM, pregestational diabetes; PR, prevalence ratio; RR, risk ratio.
(TIF)

**S16 Fig. Maternal metabolic disorders and risk of pulmonary valve stenosis in the offspring.** Obesity is defined as BMI ≥ 30 kg/m² unless other is stated; early-onset PE defined as debut before gestational week 34; PGDM are defined as DM1 or DM2; all risk estimates are adjusted unless other is stated; *, not adjusted; §, isolated defects; £, BMI ≥ 35 kg/m²; †, BMI 35-<40 kg/m²; #, untreated hypertension. Abbreviations: BMI, body mass index; DM1, diabetes type 1; DM2, diabetes type 2; GDM, gestational diabetes; OR, odds ratio; PE, preeclampsia; PGDM, pregestational diabetes; PR, prevalence ratio; RR, risk ratio.
(TIF)

**S17 Fig. Maternal metabolic disorders and risk of Ebstein's anomaly in the offspring.** Obesity is defined as BMI ≥ 30 kg/m² unless other is stated; early-onset PE defined as debut before gestational week 34; PGDM are defined as DM1 or DM2; all risk estimates are adjusted unless other is stated; *, not adjusted; §, isolated defects; £, BMI ≥ 35 kg/m²; #, untreated hypertension; $, Brodwall et al. pooled early-onset PE and severe PE. Abbreviations: BMI, body mass index; DM1, diabetes type 1; DM2, diabetes type 2; GDM, gestational diabetes; OR, odds ratio; PE, preeclampsia; PGDM, pregestational diabetes; PR, prevalence ratio; RR, risk ratio.
(TIF)

**S18 Fig. Maternal metabolic disorders and risk of septal defects in the offspring.** Obesity is defined as BMI ≥ 30 kg/m² unless other is stated; early-onset PE defined as debut before gestational week 34; PGDM are defined as DM1 or DM2; all risk estimates are adjusted unless other is stated; *, not adjusted; §, isolated defects; £, BMI ≥ 35 kg/m². Abbreviations: BMI, body mass index; DM1, diabetes type 1; DM2, diabetes type 2; GDM, gestational diabetes; OR, odds ratio; PE, preeclampsia; PGDM, pregestational diabetes; PR, prevalence ratio; RR, risk ratio.
(TIF)

**S19 Fig. Maternal metabolic disorders and risk of ventricular septal defects in the offspring.** Obesity is defined as BMI ≥ 30 kg/m2 unless other is stated; early-onset PE defined as debut before gestational week 34; PGDM are defined as DM1 or DM2; all risk estimates are adjusted unless other is stated; *, not adjusted; §, isolated defects; ¤, obesity defined from ICD-10 codes; £, BMI ≥ 35 kg/m2; €, BMI > 29 kg/m2; †, BMI 35-<40 kg/m2; #, untreated hypertension; $, Brodwall et al. pooled early-onset PE and severe PE. Abbreviations: BMI, body mass index; DM1, diabetes type 1; DM2, diabetes type 2; GDM, gestational diabetes; OR, odds ratio; PE, preeclampsia; PGDM, pregestational diabetes; PR, prevalence ratio; RR, risk ratio.
(TIF)

**S20 Fig. Maternal metabolic disorders and risk of atrial septal defects in the offspring.** Obesity is defined as BMI ≥ 30 kg/m² unless other is stated; early-onset PE defined as debut before gestational week 34; PGDM are defined as DM1 or DM2; all risk estimates are adjusted unless other is stated; *, not adjusted; §, isolated defects; ¤, obesity defined from ICD-10 codes; £, BMI ≥ 35 kg/m²; €, BMI > 29 kg/m²; †, BMI 35-<40 kg/m²; #, untreated hypertension. Abbreviations: BMI, body mass index; DM1, diabetes type 1; DM2, diabetes type 2; GDM, gestational diabetes; OR, odds ratio; PE, preeclampsia; PGDM, pregestational diabetes; PR, prevalence ratio; RR, risk ratio.
(TIF)

**S1 Table. Search protocol.** PubMed and Embase: Publications between January 1, 1990 and October 6, 2019.
(PDF)

**S2 Table. Newcastle-Ottawa quality assessment scale.**
(PDF)

**S3 Table. Characteristics of included studies.** Studies listed after number of CHD cases: Studies with most cases first etc. Abbreviations: ABDCCS, Atlanta Birth Defect Case-Control Study; BDRFSS, Birth Defects Risk Factor Surveillance Study; CHDs, congenital heart defects; CM, congenital malformations; CSL, The Consortium of Safe Labor; DM1, diabetes mellitus type 1; DM2, diabetes mellitus type 2; GDM, gestational diabetes mellitus; NBDPS, National Birth Defects Prevention Study; NOS, Newcastle Ottawa Scale; PE, preeclampsia; PGDM, pregestational diabetes.
(PDF)

**S4 Table. Newcastle-Ottawa quality assessment scale (NOS) score for each study included in the review.** The Newcastle-Ottawa Quality Assessment Scale is explained in detail in S2 Table.
(PDF)

**S1 Appendix. PRISMA checklist.**
(PDF)

## Acknowledgments

Thank you to the researchers who readily provided supplementary information. This research has been conducted using the Danish National Biobank resource supported by the Novo Nordisk Foundation.

An earlier version of the article is available at a preprint server, MedRxiv (DOI: https://doi.org/10.1101/2020.06.25.20140186).

## Author Contributions

**Conceptualization:** Gitte Hedermann, Michael Christiansen.

**Data curation:** Gitte Hedermann.

**Formal analysis:** Gitte Hedermann.

**Funding acquisition:** Gitte Hedermann, Paula L. Hedley, Ida N. Thagaard, Lone Krebs, Michael Christiansen.

**Investigation:** Gitte Hedermann, Paula L. Hedley, Ida N. Thagaard, Thorkild I. A. Sørensen, Michael Christiansen.

**Methodology:** Gitte Hedermann, Paula L. Hedley, Ida N. Thagaard, Thorkild I. A. Sørensen, Michael Christiansen.

**Project administration:** Michael Christiansen.

**Software:** Gitte Hedermann, Paula L. Hedley.

**Supervision:** Paula L. Hedley, Ida N. Thagaard, Lone Krebs, Charlotte Kvist Ekelund, Thorkild I. A. Sørensen, Michael Christiansen.

**Validation:** Gitte Hedermann.

**Visualization:** Gitte Hedermann.

**Writing – original draft:** Gitte Hedermann, Paula L. Hedley, Thorkild I. A. Sørensen, Michael Christiansen.

**Writing – review & editing:** Gitte Hedermann, Paula L. Hedley, Ida N. Thagaard, Lone Krebs, Charlotte Kvist Ekelund, Thorkild I. A. Sørensen, Michael Christiansen.

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
