## [Decision Letter · Decision Letter 0]

25 Mar 2021

PONE-D-21-00142

Maternal obesity and metabolic disorders associate with congenital heart defects in the offspring: a systematic review

PLOS ONE

Dear Dr. Hedermann,

Thank you for submitting your manuscript to PLOS ONE. After careful consideration, we feel that it has merit but does not fully meet PLOS ONE’s publication criteria as it currently stands. Therefore, we invite you to submit a revised version of the manuscript that addresses the points raised during the review process.

We look forward to receiving your revised manuscript.

Kind regards,

Linglin Xie

Academic Editor

PLOS ONE

Journal Requirements:

Reviewers' comments:

Reviewer's Responses to Questions

**Comments to the Author**

1. Is the manuscript technically sound, and do the data support the conclusions?

Reviewer #1: Partly

Reviewer #2: Yes

2. Has the statistical analysis been performed appropriately and rigorously? 

Reviewer #1: I Don't Know

Reviewer #2: Yes

3. Have the authors made all data underlying the findings in their manuscript fully available?

Reviewer #1: Yes

Reviewer #2: Yes

4. Is the manuscript presented in an intelligible fashion and written in standard English?

Reviewer #1: Yes

Reviewer #2: Yes

5. Review Comments to the Author

Reviewer #1: The article conducted a systematic review to evaluate the relationship between maternal metabolic syndrome and offspring CHD. By reviewing previous studies, the paper pointed out that some metabolic disorders could increase the risk of CHD marginally, while PGMD and early PE are strongly associated with CHD. However, some major questions are also noticed:

The article screening was conducted based on PubMed and Embase. While there could be more established information available in addition to the electronic databases such as trial registries, is there any reason to exclude those?

Why did the paper only select papers that involved cohort or case control studies that are published in English? This will reduce the comprehensiveness of the article.

As the article mentioned, the occurrence of diabetes, obesity and hypertension could be concurrent. Will there be any interaction in between when summarizing maternal metabolic disorders and different types of CHD in the offspring?

Reviewer #2: The paper is well written. the research question is well crafted and clearly stated. The authors used an appropriate tool (PRISMA) for the review. Clear definition of terms and outcome variable was done. However, the authors will need to refence many of the statements made in the background(e,g, lines 50,55,56, 60,61,62,65,66 etc)

6. PLOS authors have the option to publish the peer review history of their article (what does this mean?). If published, this will include your full peer review and any attached files.

Reviewer #1: No

Reviewer #2: **Yes: **Christopher Yilgwan

---

## [Author Response · Author response to Decision Letter 0]

26 Apr 2021

Review Comments to the Author

Reviewer #1: 

The article conducted a systematic review to evaluate the relationship between maternal metabolic syndrome and offspring CHD. By reviewing previous studies, the paper pointed out that some metabolic disorders could increase the risk of CHD marginally, while PGMD and early PE are strongly associated with CHD. However, some major questions are also noticed:

The article screening was conducted based on PubMed and Embase. While there could be more established information available in addition to the electronic databases such as trial registries, is there any reason to exclude those?

Answer: We agree that further information may be available from trial registers. We have searched the following registers with the indicated outcomes: 

WHO ICTRP (https://www.who.int/clinical-trials-registry-platform): 139 records for 121 trials, none relevant for this topic. 

BMC ISRCTN (https://www.isrctn.com/): 23 results, none relevant for this topic. 

The EU Clinical Trails Register (https://www.clinicaltrialsregister.eu/ctr-search/search): 32 results, none relevant for this topic. 

ClinicalTrials.gov: 431 results, five studies of possible relevance (ClinicalTrials.gov Identifier: NCT02914392; NCT01669057; NCT00757510; NCT00005258; NCT00005153). No one included results at ClinicalTrials.gov, none had published articles of relevance when searching PubMed using the primary investigator’s name, and therefore none of the five studies were possible to include in the review.

We have added the following in the Methods section:

“A search in a number of clinical trial registers April 12, 2021 (search term: “congenital heart defects”) identified five trials that might be relevant for this topic. However, no results were available from the trial registers or PubMed, and therefore not possible to include in the review.” (Materials and methods; Search strategy; p. 5, lines 100-103)

Why did the paper only select papers that involved cohort or case control studies that are published in English? This will reduce the comprehensiveness of the article.

Answer: We agree that limiting to English does carry the risk of overlooking data published in other languages. However, we are simply not able to evaluate studies in other languages, e.g., Chinese, Japanese, Spanish, etc. Among the studies reviewed, there were studies from various ethnicities, and we did not find noteworthy differences between these groups. Therefore, we have, as is frequently done, limited our search to English, as also mentioned as a limitation in the Discussion.

As the article mentioned, the occurrence of diabetes, obesity and hypertension could be concurrent. Will there be any interaction in between when summarizing maternal metabolic disorders and different types of CHD in the offspring?

Answer: We agree that this is a very important question and we have stressed the need of this in the conclusion of the paper. Our review shows that there is a sad shortage of studies establishing the potential effect of concurrent metabolic disorders. The small number of studies (n = 1) preclude modeling of any interactions

Reviewer #2: 

The paper is well written. the research question is well crafted and clearly stated. The authors used an appropriate tool (PRISMA) for the review. Clear definition of terms and outcome variable was done. However, the authors will need to refence many of the statements made in the background (e,g, lines 50,55,56, 60,61,62,65,66 etc)

Answer: We acknowledge the missing references and have now added more of these to the Introduction section (p. 3-4, lines 50, 55, 56, 61, 62, 66, and 80).

---

## [Decision Letter · Decision Letter 1]

14 May 2021

Maternal obesity and metabolic disorders associate with congenital heart defects in the offspring: a systematic review

PONE-D-21-00142R1

Dear Dr. Hedermann,

We’re pleased to inform you that your manuscript has been judged scientifically suitable for publication and will be formally accepted for publication once it meets all outstanding technical requirements.

Kind regards,

Linglin Xie

Academic Editor

PLOS ONE

Additional Editor Comments (optional):

Reviewers' comments:

Reviewer's Responses to Questions

**Comments to the Author**

1. If the authors have adequately addressed your comments raised in a previous round of review and you feel that this manuscript is now acceptable for publication, you may indicate that here to bypass the “Comments to the Author” section, enter your conflict of interest statement in the “Confidential to Editor” section, and submit your "Accept" recommendation.

Reviewer #1: All comments have been addressed

2. Is the manuscript technically sound, and do the data support the conclusions?

Reviewer #1: Yes

3. Has the statistical analysis been performed appropriately and rigorously? 

Reviewer #1: N/A

4. Have the authors made all data underlying the findings in their manuscript fully available?

Reviewer #1: Yes

5. Is the manuscript presented in an intelligible fashion and written in standard English?

Reviewer #1: Yes

6. Review Comments to the Author

Reviewer #1: (No Response)

7. PLOS authors have the option to publish the peer review history of their article (what does this mean?). If published, this will include your full peer review and any attached files.

Reviewer #1: No

---

## [Editor Report · Acceptance letter]

19 May 2021

PONE-D-21-00142R1 

Maternal obesity and metabolic disorders associate with congenital heart defects in the offspring: a systematic review 

Dear Dr. Hedermann:

I'm pleased to inform you that your manuscript has been deemed suitable for publication in PLOS ONE. Congratulations! Your manuscript is now with our production department. 

Kind regards, 

on behalf of

Dr. Linglin Xie 

Academic Editor

PLOS ONE